# Amygdala DEGs are associated with the immune system function: A comparative transcriptomic study of high- and low-excitability rat strains

Irina Shalaginova[1*], Marina Pavlova[2], Natalia Dyuzhikova[2]

**1** Educational and Scientific Cluster "Institute of Medicine and Life Sciences (MEDBIO)", Immanuel Kant Baltic Federal University, Kaliningrad, Russia, **2** Pavlov Institute of Physiology of the Russian Academy of Sciences, Saint-Petersburg, Russia

\* shalaginova_i@mail.ru

## Abstract

The aim of this study was to investigate differentially expressed genes (DEGs) in the amygdala of *Rattus norvegicus* with contrasting levels of nervous system excitability (high- and low-excitability). Each group consisted of 5 intact rats (n = 5). RNA sequencing was performed at on a HiSeq1500 (Illumina) generating at least 20 million paired-end reads per sample. A total of 257 DEGs were identified: 152 genes were upregulated in high-excitability rats and 105 genes up-regulated in low-excitability rats. Gene Ontology (GO) and KEGG pathway analyses revealed that the differences in gene expression were associated with immune processes such as antigen presentation and regulation of inflammation. It is also discussed, in conjunction with previous findings, that high-excitability rats may exhibit a predisposition to increased neuroinflammatory activity even without stressor exposure, potentially contributing to varying behavioral responses to stress.

## Introduction

The mechanisms underlying most psychopathologies are still not fully understood. Selective breeding of model animals offers unique opportunities to study mechanisms of diseases by creating and studying specific physiological and behavioral phenotypes. During the 1970s, two rat strains were selected at the I.P. Pavlov Institute of Physiology; they significantly differed in the excitability of their nervous system. The selection was initially aimed at the excitability threshold of the tibial nerve, but later studies showed that substantial differences in excitability were maintained also in the structures of the central nervous system of these strains [1].

Previous studies revealed that low- and high-excitability are associated with dissimilar behavior characteristics under normal conditions and at various time points after stress exposure [2,3]. There were also critical differences between the high- and low-excitability rats in molecular, genetic, cellular, and systemic studies [4]. Prior

**Data availability statement:** All relevant data are within the manuscript and its Supporting information files.

**Funding:** This research was funded by the Russian Federal Academic Leadership Program Priority 2030 at the Immanuel Kant Baltic Federal University (IKBFU) and by the State funding of the Pavlov Institute of the Physiology, Russian Academy of Sciences (N 1021062411629-7-3.1.4). The funders had no role in study design, data collection and analysis, decision to publish, or preparation of the manuscript.

**Competing interests:** The authors have declared that no competing interests exist.

research has demonstrated that rats with a genetic predisposition to high excitability of the nervous system results in an increased risk of developing a post-stress inflammatory reaction [3,5]. At the same time, changes also occurred in rats with low excitability, but they are less pronounced.

These findings underscore the role of neuroinflammatory processes in the differential stress responses observed between rat strains. Given the growing evidence that gut microbiota can influence neuroinflammation and stress responses through the gut-brain axis [6], we next examined gut microbiota composition to assess its potential contribution to these phenotypic differences. Under physiological conditions, we observed significant differences in gut microbiota composition between high- and low-excitability rats, with high-excitability rats exhibiting diminished alpha-diversity. Moreover, under stress conditions, strain-specific alterations became evident—prolonged stress increased the relative abundance of *Faecalibacterium* and *Prevotella* in high-excitability rats, while both groups exhibited a decrease in *Lactobacillus* abundance [7].

To better understand the molecular mechanisms underlying these phenotypic differences, we focused on brain transcriptomic analysis. Previous research has demonstrated that genetic and molecular regulation in the brain plays a crucial role in shaping behavioral traits and stress susceptibility [8,9]. However, the specific transcriptional differences associated with innate excitability levels remain poorly characterized. Changes in the amygdala excitability underlie behavioral disturbances characteristic of disorders such as post-traumatic stress disorder (PTSD), autism, and attention-deficit hyperactivity disorder (ADHD) [10]. Given the role of amygdala in regulating emotional and stress-related behaviors, investigating its transcriptomic profile may provide insights into the molecular mechanisms underlying the differences observed in high- and low-excitability rat strains.

The aim of the study was to identify differentially expressed genes (DEGs) in the amygdala in comparing of two rat strains with contrasting excitability of the nervous system and to determine the biological processes in which these genes are predominantly involved. The hypothesis of the study was that the transcriptome of the amygdala in intact rats from the two strains with contrasting excitability differs, and these differences are primarily related to the molecular mechanisms regulating neuronal excitability.

## Materials and methods

### Animals

The study involved 5-month-old male rats from two strains (n = 5 in each group) selectively bred for nervous system excitability at the Pavlov Institute of Physiology (Russian Academy of Sciences). HT - (high excitability threshold, low excitability) and LT - (low excitability threshold, high excitability), were developed to investigate differences in nervous system reactivity. The strains are part of the biological collection at the Pavlov Institute of Physiology, RAS (NoGZ 0134-2018-0003), with patents for selective breeding (No 10769 and 10768).

The original stock was derived from Wistar rats. Selection was based on the neuromuscular excitability threshold in the tibial nerve stimulation test (rectangular electrical impulses of 2 ms duration). The first two generations were bred using full siblings, while subsequent generations were bred randomly within the line. By the 10th generation, the breeding stabilized, resulting in a fourfold difference between the strains, which significantly exceeded the variability within each strain. Experiment included 10 intact animals (n = 5 each strain).

All animals were housed under standard conditions (23 ± 2°C; 12 h/12 h light/dark cycle) with free access to food and water in the vivarium of the Pavlov Institute of Physiology, RAS. All animals in the study were housed in the same room in the vivarium three per cage, until reaching five months of age, after which they were decapitated in an intact state.

Decapitation was performed using a guillotine without anesthesia. The choice to avoid anesthesia was based on evidence that isoflurane and other anesthetics can rapidly alter cytokine gene expression [11,12], which could compromise the study's focus on immune gene expression. Therefore, anesthesia was deemed unsuitable for this experimental design.

The decapitation procedure was carried out by a specially trained staff member with the necessary skills in humane euthanasia techniques.

All experiments were conducted in accordance with the European Community Council Directive (86/609/EEC) on the use of animals for experimental re-search. The protocol was approved by the Institutional Animal Care and Use Committee of the Pavlov Institute of Physiology, RAS (Protocol No. 09/16, dated 16.09.21).

## Tissue collection, RNA extraction and sequencing

To analyze the transcriptomes of the amygdala in two rat strains tissue samples were collected and total RNA was extracted using Trisol reagent and the PureLink RNA micro Kit (Invitrogen, Carlsbad, CA, USA). Sequencing was performed at the Genoanalitika Lab, Moscow, Russia. Libraries for sequencing were prepared using the NEBNext® Ultra™ II RNA Library Prep Kit (NEB). Sequencing was done on a HiSeq1500 (Illumina), generating at least 20 million single-end 50-nucleotide reads per sample. Read mapping and counting were performed with *STAR* 2.7.9a [13]. All parameters were used by default, with the exception of *outFilterMismatchNmax 3*, which limited the maximum number of read mismatches to three.

The reference genome used for alignment was the *Rnor_6.0 assembly*, and annotation data were obtained from *Ensembl v.99*. Read counting was performed directly in *STAR* [13] using the *quantMode GeneCounts* option, which provided raw gene-level expression values.

Differential expression analysis was conducted using *DESeq2* (v.1.28.1) [14] in the R environment. Raw read count data obtained from *STAR* were imported into R. Normalization was performed using the *estimateSizeFactors()* function, which applies the median-of-ratios method. Dispersion modeling was carried out using the *estimateDispersions()* function. Differentially expressed genes were identified using the *DESeq()* function, which is based on a negative binomial distribution model. Condition comparisons were performed using the *Wald* test, and multiple testing correction was applied using the Benjamini-Hochberg method to control the false discovery rate. Quality of sequencing see in data S2 Table.

## RNA sequencing analyses

A volcano plot was generated to visualize interstrain differential gene expression using R and RStudio 2024.04.2. In our work, upregulated genes were genes with higher expression in the LT strain compared to the HT strain (log2FoldChange > 0.38, padj < 0.05), while downregulated genes were genes with lower expression in the LT strain compared to the HT strain (log2FoldChange < -0.38, padj < 0.05). The plot was created with *ggplot2* [15], and interactive volcano plot using *plotly* [16].

Gene Ontology (GO) enrichment analysis was performed using the Database for Annotation, Visualization and Integrated Discovery (DAVID) 6.8 v2021q4 (https://david.ncifcrf.gov/). to find the overrepresented biological processes,

molecular functions, and cellular components associated with DEGs obtained from our RNA-seq experiment. Significance was assessed through the *clusterProfiler* [17] package in R and *org.Rn.e.g.,db* annotation database specific to *Rattus norvegicus*. For the GO analysis, up- and down-regulated genes were treated as a single list. The reference set was a background list of all the expressed genes from our RNA-seq experiment. Multiple testing correction was performed using the Benjamini-Hochberg (BH) method, with a p-value cutoff of 0.05 and a q-value cutoff of 0.2.

KEGG pathway enrichment analysis was performed using the *enrichKEGG* function from the *clusterProfiler* package in R and *mapIds* function from the *org.Rn.e.g.,db* for conversion from Ensembl IDs to Entrez Gene IDs. Any non- mapped genes were excluded from the analysis. Cutoff for p-value: 0.05, Benjamini-Hochberg (BH) correction for multiple testing.

STRING database (version 12.0) was used to analyze a protein-protein interaction (PPI) network, confidence score threshold - 0.4. Direct and indirect protein interactions, integrating data from curated databases, experimental studies, and computational predictions were used. Clusters were identified using the k-means algorithm within STRING [18]. For the construction of the protein-protein interaction (PPI) network, we identify DEGs associated with immune and inflammatory processes, we filtered GO enrichment results for BP terms containing the keywords "inflamm", "immune", or "complement" using regular expressions in R. Genes linked to the filtered GO terms were extracted, deduplicated, and saved as a non-redundant list for further analysis.

## Results

To determine the gene expression profile specific to the amygdala in rats with high and low levels of nervous system excitability (n = 5/group), total RNA-sequencing (RNA-seq) was performed. mRNA levels in high excitable (LT) rats were compared to the corresponding expression levels in the low excitable strain (HT), 257 genes were recognized as differentially expressed; 152 were upregulated (genes with higher expression in the LT strain compared to the HT strain) and 105 were downregulated (genes with lower expression in the LT strain compared to the HT strain) (Fig 1, Table in S1 Table).

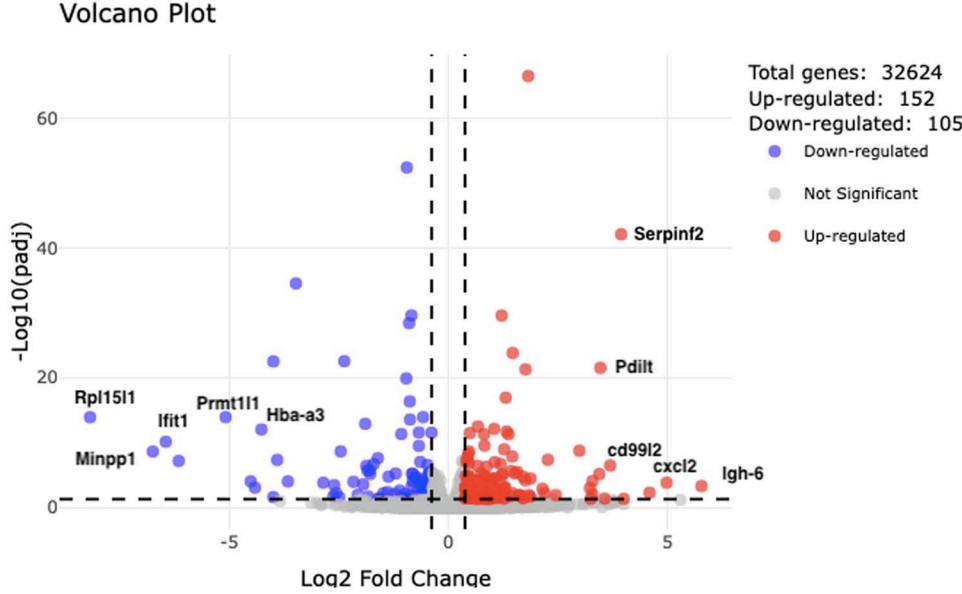

**Fig 1. Volcano plot of differentially expressed genes (DEGs) in the amygdala of two rat strains with different levels of nervous system excitability.** LT – low threshold, high excitable; HT – high threshold, low excitable; Cut off: log2fold change ≥ |0.38| and padj ≤0.05 (FDR).

Among the genes with higher expression in the LT strain compared to the HT strain, *Serpinf2*, *Pdlit*, *Cd99l2*, *Cxcl2*, and *Igh-6* were notably identified. A top-5 of significantly genes with lower expression in the LT strain compared to the HT strain include *Rpl15l*, *Ifit1*, *Prmt1*, *Minpp1*, and *Hba-a3* (Table 1).

The hierarchical clustering of the top 50 differentially expressed genes (DEGs) in the amygdala reveals a clear distinction in expression profiles between LT and HT rats, highlighting distinct transcriptional patterns between the groups (Fig 2).

GO enrichment analysis for the biological process (BP) category shows that DEGs were significantly overrepresented in immune-related processes (Fig. 3). The most statistically significant processes include "regulation of adaptive immune response," "antigen processing and presentation of peptide antigen via MHC class II," and "regulation of viral entry into host cell.

To clarify the interactions and functional relationships among the differentially expressed genes, a protein-protein interaction (PPI) network was constructed (Fig 7).

The resulting network was divided into distinct clusters, each representing a different aspect of the immune response. First cluster (Red): contain 8 proteins which primarily involved in the "Antigen processing and presentation of exogenous peptide antigen." Second cluster (Yellow): 7 DEGs products associated with the "Cellular response to interferon-beta," highlighting their role in antiviral defense mechanisms. Green cluster: includes 4 proteins related to the "Regulation of complement activation," which plays a crucial role in innate immunity and inflammation. Blue cluster: 2 proteins, Cxcl2 and

**Table 1. Description and functions of TOP-10 DEGs in the amygdala of high-excitable (LT) rats compared to low-excitable (HT) rats.**

| Gene | Expression in LT strain compared to the HT strain | Localization | Description | Functions |
|------|---------------------------------------------------|--------------|-------------|-----------|
| *Serpinf2* | Higher-expressed gene | Extracellular matrix | Serpin Family F Member 2 | The protein is a major inhibitor of plasmin, which degrades fibrin and various other proteins. |
| *Pdilt* | Higher-expressed gene | Endoplasmic reticulum | Protein disulfide isomerase-like, testis expressed. | Is involved in protein folding and acts as a chaperone. Is active in germ cell migration and spermatid development; the function in the brain has not been studied. |
| *Cd99l2* | Higher-expressed gene | Extracellular region, Cell membrane | CD99 molecule-like 2 | Plays a role in a late step of leukocyte extravasation helping cells to overcome the endothelial basement membrane. |
| *Cxcl2* | Higher-expressed gene | Extracellular space | Chemokine (C-X-C motif) ligand 2, a mediator of inflammation. | A part of a chemokine superfamily that encodes secreted proteins involved in immunoregulatory and inflammatory processes. |
| *Igh-6* | Higher-expressed gene | Plasma cells | Immunoglobulin heavy chain 6 | Involved in immunoglobulin production. |
| *Rpl15l* | Lower-expressed gene | Ribosome | Large Ribosomal Subunit Protein EL15 | Encodes a member of the L15E family of ribosomal proteins and a component of the 60S subunit. |
| *Ifit1* | Lower-expressed gene | Cytoplasm | Interferon-induced protein with tetratricopeptide repeats 1. | Specifically binds single-stranded RNA, thereby acting as a sensor of viral single-stranded RNAs and inhibiting expression of viral messenger RNA. |
| *Prmt1* | Lower-expressed gene | Nucleus | Member of the protein arginine N-methyltransferase (PRMT) family. | Post-translational modification of target proteins. |
| *Minpp1* | Lower-expressed gene | Endoplasmic reticulum | Multiple inositol polyphosphate phosphatase 1 | An enzyme that removes 3-phosphate from inositol phosphate substrates. May control the availability of intracellular calcium and iron, which are important for proper neuronal development and homeostasis [19]. |
| *Hba-a3* | Lower-expressed gene | Extracellular space | hemoglobin alpha, adult chain 3. | Enables heme binding activity and organic acid binding activity. Predicted to be involved in hydrogen peroxide catabolic process; positive regulation of cell death; and response to hydrogen peroxide. |

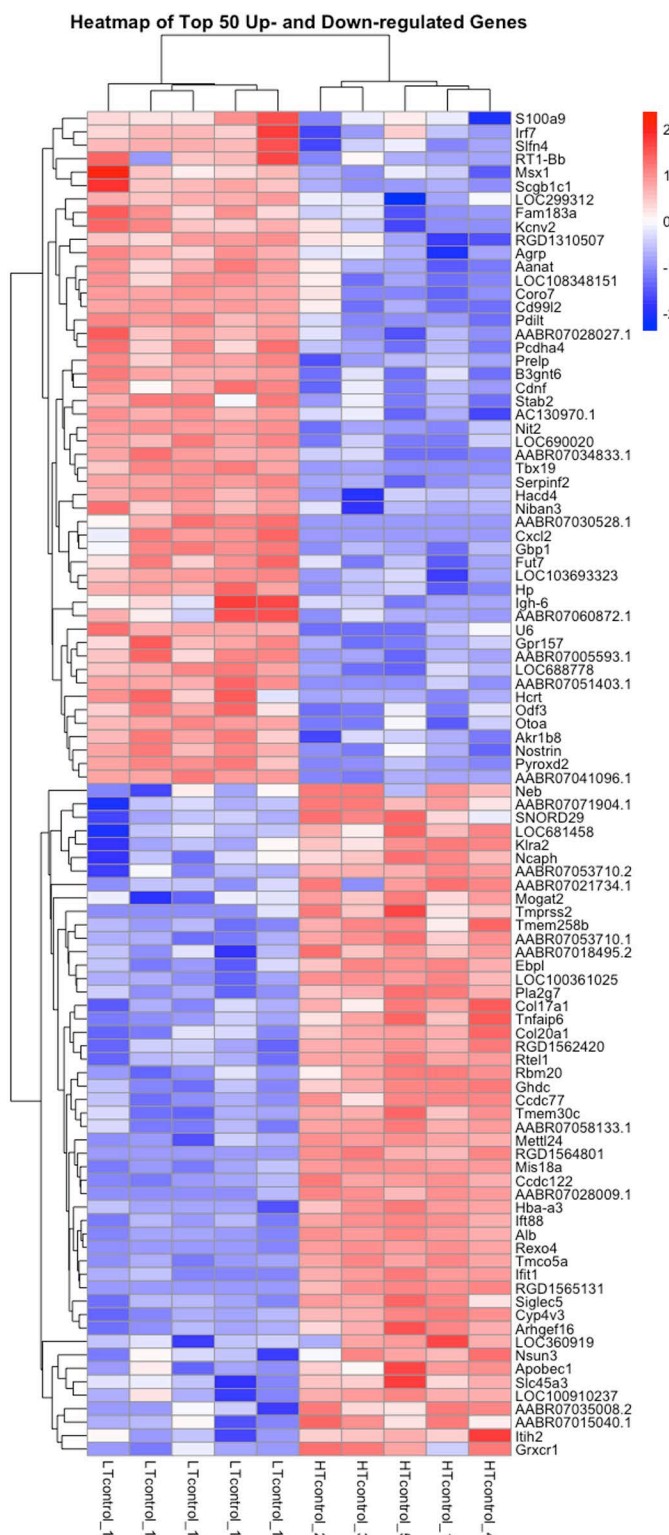

**Fig 2. Heatmap of top-50 differentially expressed genes in the amygdala of LT vs. HT rat strains.** The color scale represents normalized expression levels, with blue indicating lower ex-pression and red indicating higher expression. Cut off: log2fold change ≥ |0.38| and padj ≤0.05 (FDR). To explore the biological significance of the differentially expressed genes, GO enrichment analysis was performed (Figs 3–5).

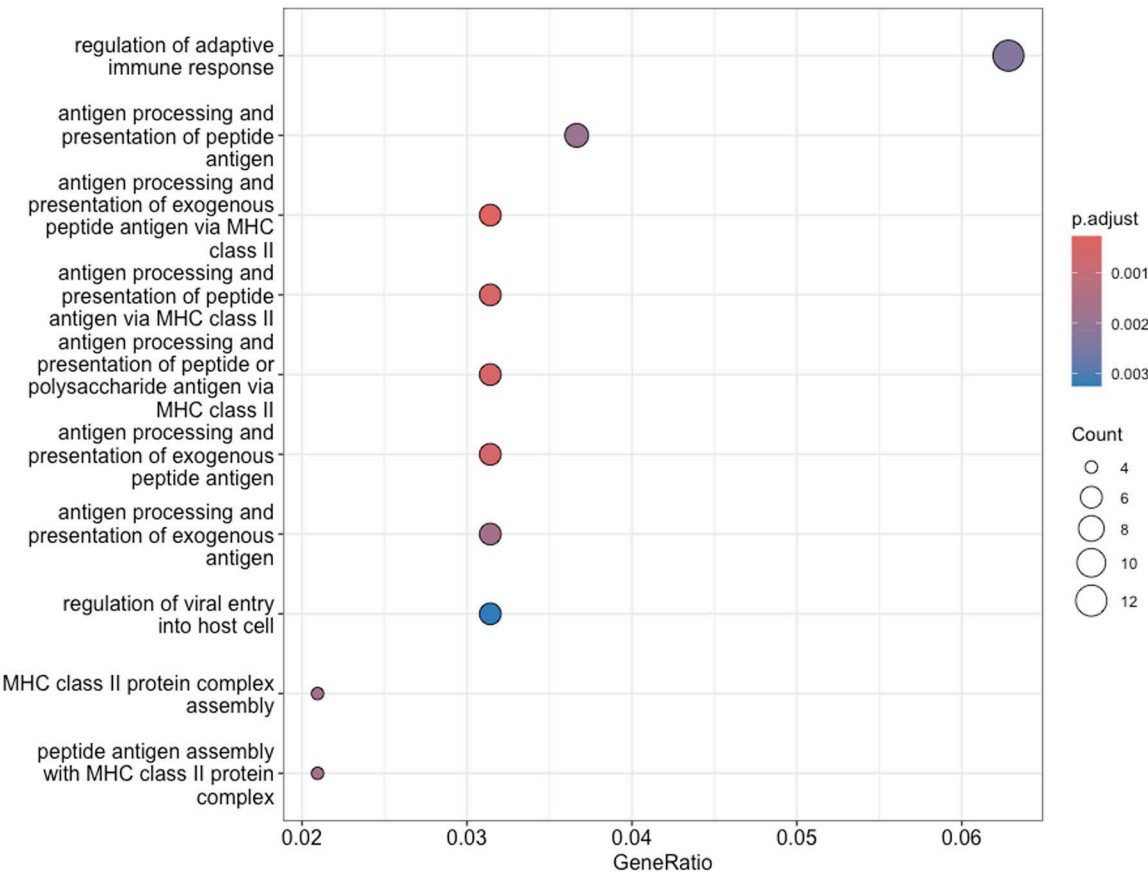

**Fig 3. Gene ontology (GO) enrichment analysis of biological processes (bp) associated with differentially expressed genes.** The x-axis represents the GeneRatio, which indicates the proportion of DEGs associated with each GO term relative to the total number of DEGs. The color gradient of the dots corresponds to the adjusted p-value (padj), while the size of the dots reflects the number of genes associated with each term. The same captions are applied in Figs 4 and 5.

S100a9 which involved in processes related to inflammation and immune cell recruitment. These clusters are visualized in Fig 7. The detailed descriptions of the genes with the highest number of interactions, including their known functions and involvement in immune processes, are provided in Table 2.

These findings highlight potential molecular differences between LT and HT rats, particularly in immune-associated processes and pathways.

## Discussion

Our findings reveal significant differences in gene expression between the high excitability (LT) and low excitability (HT) rat strains in a normal physiological state. One of the key findings of this study is the higher expression of genes involved in immune responses in LT rats compared to HT rats. For example, the upregulated genes with the highest fold change, such as *Cxcl2* and *Igh-6*, are well-known mediators of inflammation and immune cell recruitment [20]. Research has shown that *Igh-6* is expressed in the brain, but its protein structure differs from that of B-cell immunoglobulins [21]. Neural-specific immunoglobulins have been shown to perform functions beyond classical immune responses. Previous studies have demonstrated that *Igh-6* knockout leads to astrocyte de-differentiation into neural progenitor cells via the BMP/YAP1/TEAD3 pathway, however increased *Igh-6* expression activates the CRTC1-CREB-BDNF pathway, which

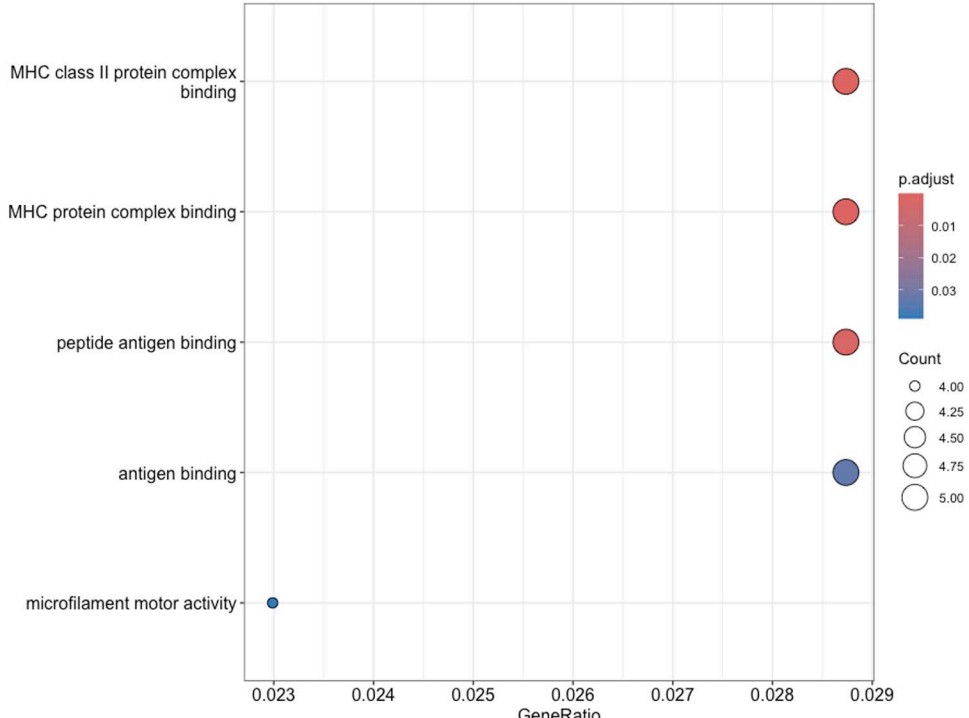

**Fig 4. Gene ontology (GO) enrichment analysis of molecular functions (MF) associated with differentially expressed genes.** GO analysis of molecular functions (MF) also revealed significant overrepresentation in functions related to immune response (Fig 4). Key molecular functions also involved in antigen processing and presentation: "MHC class II protein complex binding, "MHC protein complex binding", and "peptide antigen binding". Additionally, "microfilament motor activity" was also enriched, indicating potential involvement in cytoskeletal dynamics. The top enriched cellular components (Fig 5) include the "MHC class II protein complex" and "MHC protein complex".

plays a crucial role in gliogenesis and astrocyte maintenance [22]. Taken together, these findings suggest that *Igh-6* contributes to the regulation of glial stability and neural plasticity.

CD99 molecule-like 2 (CD99L2) is a membrane protein with an extracellular domain of 237 amino acids, which is highly conserved between mouse and human. It plays a crucial role in leukocyte infiltration into the central nervous system (CNS) during autoimmune diseases such as multiple sclerosis and its animal model [23]. The entry of leukocytes through the blood-brain barrier is a key step in the development of inflammatory brain diseases. Prior research has shown, that inactivation of *cd99l2* significantly reduced the number of leukocytes entering the CNS, which alleviated the symptoms of the disease [23]. Neuronal function of *cd99l2* has only recently been characterized [24]. CD99L2 is primarily expressed in neurons, where it regulates neurite outgrowth and the development of excitatory synapses. It also modulates the expression of immediate-early genes, including *Arc*, *Egr1*, and *c-Fos*, through the inhibition of CREB and SRF transcriptional activity. CD99L2 knockout mice exhibit impaired excitatory synaptic transmission and plasticity, leading to deficits in spatial memory and contextual fear conditioning. We observed higher *cd99l2* expression in LT rats compared to HT rats, suggesting a possible link between this gene and differences in excitability. Given its dual role in immune cell trafficking and neuronal function, *cd99l2* may contribute to both the heightened excitability and the increased immune-related gene expression observed in LT rats.

Another gene from the top 5 upregulated in the high-excitable strain compared to the low-excitable is *serpinf2*. Serpins are protease inhibitors, and although *serpinf2* is primarily studied in the context of fibrinolysis and the blood coagulation system, all the components of plasmin cascade are present in the brain [25]. However, its functions in the brain remain

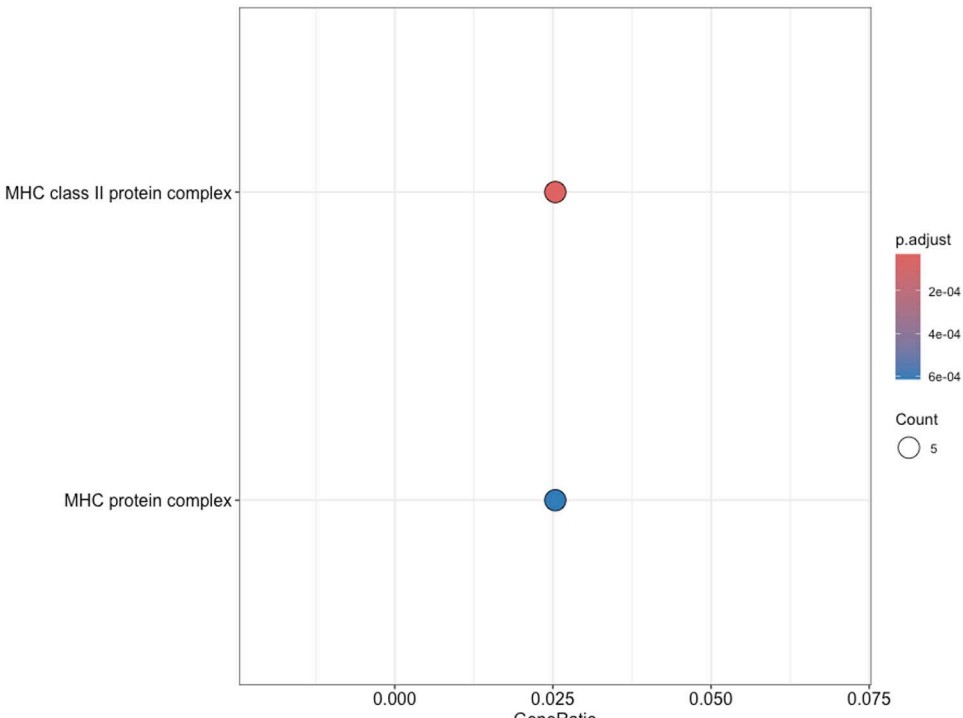

**Fig 5. Gene ontology (GO) enrichment analysis of cellular components (CC) associated with differentially expressed genes.** Additional KEGG pathway enrichment analysis revealed significant enrichment in several immune-related pathways (Fig 6). Pathways such as "Antigen processing and presentation", "Viral myocarditis", and "Phagosome" were enriched.

unclear. There is evidence that the dysregulation of serpins is involved in Alzheimer's disease and prion diseases [26,27]; for instance, in mice infected with prions, elevated levels of SerpinF2 were observed compared to controls [25].

Expression of *Rpl15l* - Large Ribosomal Subunit Protein EL15 was decreased in our study in LT compare to HT. This fact may reflect a redistribution of transcription resources, which is similar to what was seen in the study of picrotoxin-induced neuronal activation (PTX) [28]. Was found that several genes responsible for encoding ribosomal proteins, which typically have high levels of baseline expression, were downregulated shortly after neuronal stimulation. This downregulation was understood as a shift in transcription resources towards genes that drive activity and are crucial for neuronal plasticity. In a similar way, the observed decrease in *Rpl15l* mRNA levels in high-excitability (LT) rats, even without any external stimuli, might reflect an adaptive process where the brain reallocates resources to maintain consistently high levels of neuronal activity.

If the downregulation of ribosomal genes in the amygdala is sustained over time, it could result in significant disruptions to cellular processes and brain function. Research shows that a deficiency in ribosomal proteins can disrupt the expression of genes across several critical functional categories, including the cell cycle, cellular metabolism, signal transduction, and development [29,30]. This suggests that chronic downregulation might impair essential cellular processes, potentially affecting overall brain function.

Expression of *Ifit1* also was decreased in LT compare to HT, it is known, that *Ifit1* coordinates antiviral and antibacterial immune response by activating the TLR4 pathway when faced with chemical signals like lipopolysaccharides (LPS). Silencing of *Ifit1*, in mouse macrophages can weaken the immune system's ability to activate key downstream genes, leading to a diminished immune response [31]. This reduced antiviral defense could push the immune system to overreact

**Table 2. Description and functions of the DEGs products, with the highest node degrees associated with immune system.**

| Protein | Node Degree | Expression of relevant gene in LT strain compared to the HT strain | Description | Functions |
|---|---|---|---|---|
| B2m + | 9 | Higher-expressed gene | Beta-2-microglobulin, a component of MHC class I molecules, crucial for the surface expression of these molecules. | Antigen presentation, immune response, T-cell activation. |
| Cd74 + | 9 | Higher-expressed gene | CD74 (Invariant chain), a critical component of the MHC class II complex, involved in the presentation of antigens to the immune system. | Antigen processing and presentation, immune response, T-cell activation. |
| Irf7 + | 7 | Higher-expressed gene | Interferon regulatory factor 7, a transcription factor that plays a key role in the innate immune response to viral infections. | Regulation of interferon production, antiviral defense, immune response. |
| RT1-Bb + | 6 | Higher-expressed gene | Part of the MHC class II molecule, involved in presenting extracellular antigens to CD4+T cells. | Antigen presentation, immune response, activation of helper T cells. |
| Ifit1 -- | 6 | Lower-expressed gene | Interferon-induced protein with tetratricopeptide repeats 1, involved in antiviral responses, particularly in the recognition of viral RNA. | Antiviral response, immune modulation, interferon response. |
| Ifit3 + | 6 | Higher-expressed gene | Interferon-induced protein with tetratricopeptide repeats 3, which works in tandem with Ifit1 in antiviral responses. | Antiviral response, regulation of immune signaling. |
| Tlr3 + | 5 | Higher-expressed gene | Toll-like receptor 3, recognizes double-stranded RNA, typically from viruses, and triggers an immune response. | Pathogen recognition, activation of the innate immune system, antiviral defense. |
| RT1-Db1 + | 5 | Higher-expressed gene | Part of the MHC class II molecule, involved in presenting extracellular antigens to CD4+T cells. | Antigen presentation, immune response, activation of helper T cells. |
| RT1-Da + | 5 | Higher-expressed gene | Another component of the MHC class II complex, playing a similar role to RT1-Bb and RT1-Db1 in antigen presentation. | Antigen presentation, immune response, activation of helper T cells. |
| Fcgr2b+ | 5 | Higher-expressed gene | Fc gamma receptor IIb, a receptor for the Fc region of IgG antibodies, playing a role in modulating immune responses. | Regulation of immune response, inhibition of antibody production, control of inflammation. |

in other ways. In high-excitability rats, where *Ifit1* is less active, this imbalance might result in a compensatory chronic state of inflammation.

It has been clearly demonstrated that dysregulation of PRMT-family enzymes leads to neurological consequences [32,33]. As an example, a wide range of pathologies are attributed to downregulation of PRMT1, PRMT4, PRMT5, PRMT6, PRMT7, and PRMT8 in the brain, including increased neuronal hyperexcitability, astroglial loss, and failure of synaptic plasticity [for review Couto]. In the high-excitability rats, observed in our study the down-regulation of *Prmt1* may similarly contribute to heightened neuronal excitability seen with PRMT7. It has been demonstrated that arginine methylation by PRMT7 regulates sodium leak channel NALCN [34]. NALCN regulates the resting membrane potential in neurons [35]; thus, its activity directly impacts cell excitability. In hippocampal neurons from PRMT7-deficient mice, was found greater cellular excitability, a depolarized resting potential, and, compared with wild-type neurons, lower activation thresholds [36]. This study revealed that PRMT7 specifically methylates one arginine residue, Arg1653, of NALCN, which affects channel sensitivity to calcium signals. Loss of PRMT7 function reduces the methylation of this residue, decreases inhibition of NALCN, and increases neuronal excitability. It means that the downregulation of *Prmt1* in a rat high-excitable strain could interfere with crucial post-translational modifications that are required to maintain neural stability, probably leading to a state of hyperexcitability.

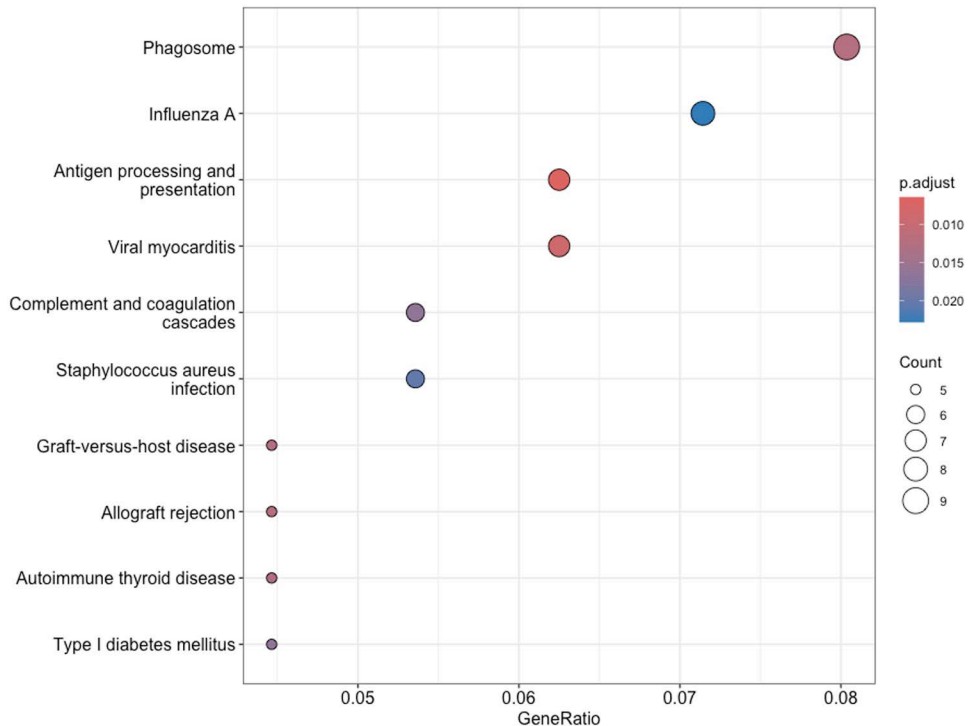

**Fig 6. KEGG pathway enrichment analysis among the differentially expressed genes in LT vs HT rat strain.**

*Minpp1* gene in high-excitability rats was also lower-expressed in LT compare to HT. Research suggests that MINPP1 is critical for the metabolism of inositol polyphosphates – a key molecules involved in various cellular processes. Decreasing of MINPP1 leads to the accumulation of inositol hexaphosphate (IP6), which binds intracellular ions like calcium and iron, reducing their availability to cells [19]. This imbalance can negatively affect neuronal differentiation and increase cell death, contributing to neurodegenerative processes. Given these findings, the reduced expression of *Minpp1* gene in high-excitable rats might contribute to increased neuronal excitability and a predisposition to neurological disorders, making this gene an important focus for further study.

We also found among the top-5 lower-expressed genes in LT rats compare to HT gene *Hba-a3* that enables heme binding activity and organic acid binding activity. Most often, this gene is discussed to be associated with oxygen transport, but research studies have investigated its involvement in the context of Alzheimer's disease [37]. It was found that it binds to amyloid-beta (Abeta) [38]. This binding could influence the accumulation or clearance of Abeta in the brain, potentially damage neurons observed in Alzheimer's disease. Within such a framework, the downregulation of *Hba-a3* in high-excitable rats might have a similar impact and might influence the functional state of the neurons.

The GO enrichment analysis of the DEGs in the amygdala of LT and HT rat strains showed an absolute predominance of significant enrichment in immune-related pathways in biological processes, molecular functions, and cellular components. For instance, processes such as "regulation of adaptive immune response", "antigen processing and presentation of peptide antigen via MHC class II", and "regulation of viral entry into host cell" were significantly overrepresented. This suggests that genes involved in immune function significantly engaged in the differences in transcriptomic profiles of the amygdala of high-excitable (LT) and low-excitable (HT) rat strains. This may possibly reflect the specific role of the amygdala, which is more closely associated with the immune-related processes then another brain structure involved in stress reaction and emotional regulation.

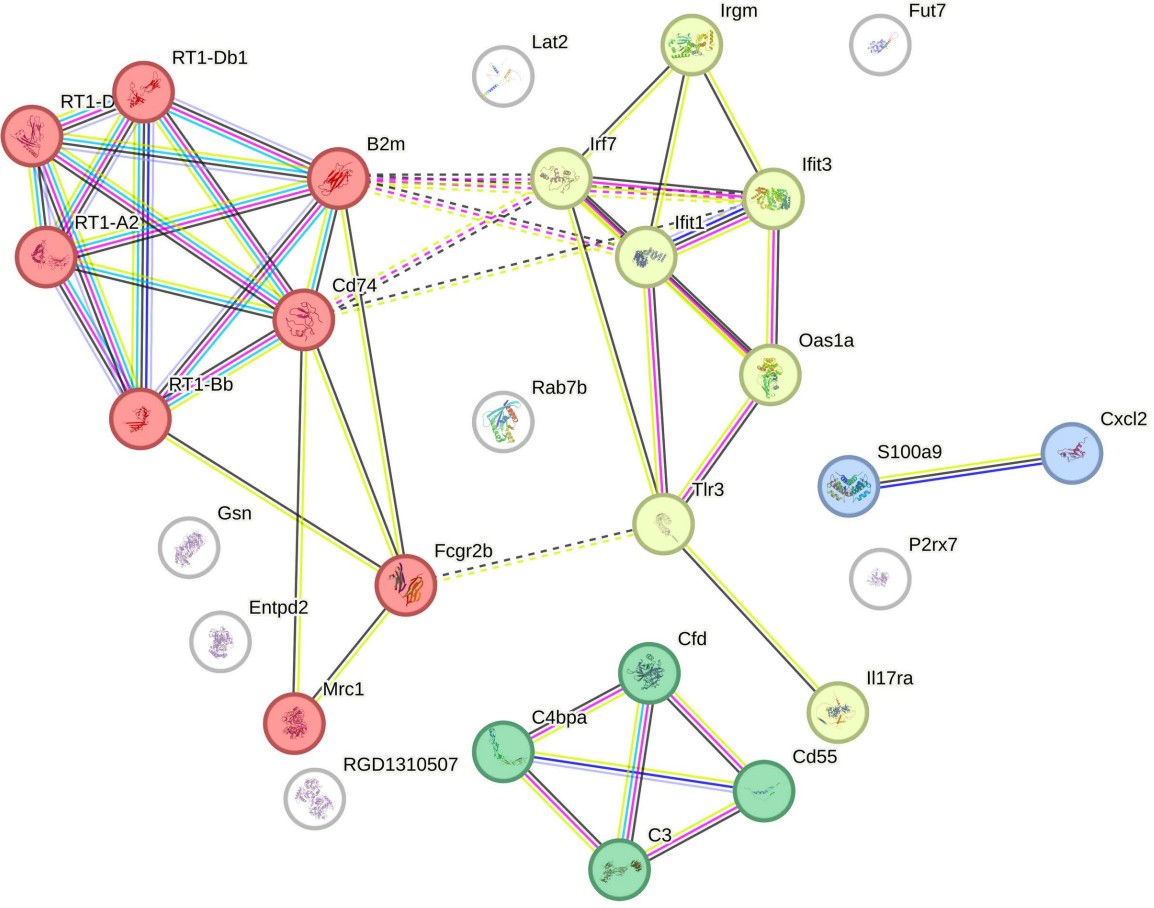

**Fig 7. Protein-protein interactions of products encoded by DEGs identified by comparing rat strains with different excitability of the nervous system.** Colored nodes: Query proteins and their first-level interactors; White nodes: Second-level interactors; Filled nodes: Proteins with known or predicted 3D structure; Light blue and pink edges: Known interactions from databases and experiments; Green, red, and blue edges: Predicted interactions based on gene neighborhood, fusions, and co-occurrence; Yellow, black, and purple edges: Other associations based on textmining, co-expression, and protein homology; Solid edges: known protein-protein interactions; Dashed edges: predicted protein-protein interactions.

The KEGG Pathway Enrichment Analysis also reveals significant immune-related activity in intact animals, highlighting pathways such as "antigen processing and presentation" and "viral response mechanisms". Despite the absence of external stressors or infection, the activation of these pathways suggests an inherent immune readiness or basal inflammatory state, possibly driven by internal genetic factors. This intrinsic immune activity could underlie the observed physiological and behavioural differences between high- and low-excitability rat strains, indicating that even without external triggers, their immune systems are primed, potentially contributing to their distinct stress responses.

Our previous studies demonstrated notable differences in post-stress neuroinflammation between high- and low-excitable rat strains: after the chronic stress exposure LT rats exhibiting increasing in number of microglial cells [3] and elevated mRNA level of pro-inflammatory cytokines like IL-1β in the brain including amygdala [5]. In contrast, low-excitable HT rats had a delayed and milder inflammatory response, coupled with a more dynamic microbiota [7] that might contribute to their enhanced stress resilience. The more diverse and dynamic microbiota in low-excitable rats might help regulate the immune response, potentially through the production of metabolites like short-chain fatty acids that influence gene expression and inflammatory pathways activation. Also, variations in genes related to immune regulation,

neuroinflammation, or stress response could predispose one strain to heightened sensitivity to neuroinflammation and stress disorders.

It is known that cytokines, including IL-1β, play the critical role in modulating the activity of the amygdala, particularly in the context of stress and anxiety-related behaviours [for review 39]. The comparison of facts that amygdala is involve in emotional regulation and its sensitivity to inflammatory signals suggest that the differential expression of immune-related genes in our rat models could influence amygdala function, contributing to the behavioural differences observed between the strains [3]. The elevated cytokine levels and immune gene activity seen in the high-excitable rats may be driving an overactivation of amygdala, thereby enhancing neuroinflammatory processes and potentially leading to more severe stress-related behavioural abnormalities.

Moreover, these findings underline the importance of considering the genetic context of these rats, as their predisposition to a stronger immune response could be rooted in specific genetic variations that regulate cytokine expression and immune activation. To better understand these mechanisms, it would be valuable to conduct a comprehensive genotyping of the rat strains. This could help identify genetic factors that contribute to the differential expression of immune-related genes, offering further insights into how genetic background influences stress resilience or susceptibility through the modulation of immune responses within the brain.

STRING analysis shows that proteins encoded by DEGs are involved in various immune processes and demonstrate significant interconnection, forming potential net-works. Although these results reflect general interaction trends rather than specific data on our rat strains, this may suggest that immune responses in high-excitable rats may be closely coordinated. However, this requires further study in relation to this particular rat strains.

One of the identified clusters (red in Fig 7) highlights the crucial role of antigen presentation, specifically through the MHC pathway. These processes are essential for initiating adaptive immunity, in recognizing and responding to pathogens or abnormal own cells [40]. The dysregulation of genes in this cluster suggests a heightened state of immune regulation, which could predispose high-excitability rats to a more severe immune response.

Cluster "Cellular Response to Interferon-Beta" (yellow in Fig 7) involves genes related to interferon-beta response, a this is a key part of cytokine in antiviral defense [41]. The presence of this cluster indicates that even in intact high-excitable animals may have a primed state of antiviral defense, which could be a protective mechanism but also a reason of chronic immune activation [36].

"Regulation of Complement Activation" cluster (green in Fig 7) is involved in the complement system, which plays a crucial role in innate immunity by marking pathogens for destruction and activating inflammation. It is known that dysregulation of complement activation can lead to autoimmune diseases and chronic inflammation [42]. The enrichment of this cluster suggests a potential predisposition to upregulation not only adaptive but innate immune responses, which could shift homeostasis to the neuroinflammation in the amygdala of these rats.

Blue cluster – "Inflammation and Immune Cell Recruitment" the smallest but also are often involved in chronic inflammatory responses and play a critical role in the recruitment of neutrophils and subsequent inflammatory responses in the brain [43,44] and have been associated with various inflammatory diseases and microglial activation [45,46].

In recent years researches has pointed to a link between immune responses and neural circuits, especially in mood disorders such as severe depressive disorder (MDD) and anxiety [47,48]. Inflammation can affect areas of the brain involved in the regulation of emotions, such as the amygdala, which plays an important role in the processing of fear and anxiety. This relationship between immune activation and neural response, although well documented in human conditions such as MDD [49], can also extend to animal models, which makes it possible to study the underlying mechanisms from an evolutionary point of view. The observed correlation between increased inflammation and activation of the amygdala in patients with MDD may be somewhat similar to the immune and neuroinflammatory processes observed in highly excitable rats. Hyperactivation of the amygdala, potentially caused by underlying inflammatory mechanisms, may be a common feature of different species and contributes to anxiety-related behaviours.

In conclusion, the interplay between genetic predisposition, immune system activity, and neuroinflammation is likely a key factor in the differing stress responses observed in high- and low-excitability rat strains. These findings not only clarify our understanding of the neuroimmune mechanisms underlying stress-related behaviours but also highlight potential targets for therapeutic intervention aimed at modulating immune responses to improve stress resilience.

## Conclusion

Thus, analysis of the amygdala transcriptome of LT and HT rats revealed 257 genes that change expression in response to selection by contrast (high and low) levels of excitability of the nervous system. Differentially expressed amygdala genes are predominantly involved in processes associated with immune responses and may be associated with a predisposition to the development of neuroinflammation in post-stress pathology of anxiety spectrum behavior.

In the future, sequencing of the transcriptome (detection of DEGs, functional annotation, and determination of biological pathways) in the amygdala and other parts of the brain immediately after psychoemotional stress could reveal the neurogenomic mechanisms involved in the development of pathological conditions associated with the level of hereditarily determined excitability of the nervous system, potentially identifying targets for their prevention and correction.

## Supporting information

**S1 Table. The table with filtered DEGs.**
(XLS)

**S2 Table. Quality of sequencing.**
(CSV)

## Acknowledgments

We would like to thank Genoanalytica Lab, Moscow, Russia, where the samples were sequenced.

## Author contributions

**Conceptualization:** Irina Shalaginova, Natalia Dyuzhikova.

**Data curation:** Marina Pavlova.

**Formal analysis:** Irina Shalaginova, Marina Pavlova.

**Funding acquisition:** Irina Shalaginova, Natalia Dyuzhikova.

**Investigation:** Irina Shalaginova.

**Methodology:** Marina Pavlova, Natalia Dyuzhikova.

**Project administration:** Irina Shalaginova, Natalia Dyuzhikova.

**Visualization:** Irina Shalaginova.

**Writing – original draft:** Irina Shalaginova.

**Writing – review & editing:** Marina Pavlova, Natalia Dyuzhikova.

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
