## [Decision Letter · Decision Letter 0]

21 Feb 2025

PONE-D-24-49620Аmygdala DEGs are associated with the immune system function: a comparative transcriptomic study of high- and low-excitable rat strainsPLOS ONE

Dear Dr. Shalaginova,

Thank you for submitting your manuscript to PLOS ONE. After careful consideration, we feel that it has merit but does not fully meet PLOS ONE’s publication criteria as it currently stands. Therefore, we invite you to submit a revised version of the manuscript that addresses the points raised during the review process. After careful consideration by 2 Reviewers and an Academic Editor, all of the critiques of the Reviewers must be addressed in detail in a revision to determine publication status. If you are prepared to undertake the work required, I would be pleased to reconsider my decision, but revision of the original submission without directly addressing the critiques of the Reviewers does not guarantee acceptance for publication in PLOS ONE. If the authors do not feel that the queries can be addressed, please consider submitting to another publication medium. A revised submission will be sent out for re-review. The authors are urged to have the manuscript given a hard copyedit for syntax and grammar.

We look forward to receiving your revised manuscript.

Kind regards,

Stephen D. Ginsberg, Ph.D.

Section Editor

PLOS ONE

Journal Requirements:

1) Russian Federal Academic Leadership Program Priority 2030 at the Immanuel Kant Baltic Federal University (IKBFU)

2) State Program 47 SP «Scientific and technological development of the RF», topic 0134-2019-0002 (Pavlov Institute of Physiology, Russian Academy of Sciences).

Reviewers' comments:

Reviewer's Responses to Questions

**Comments to the Author**

1. Is the manuscript technically sound, and do the data support the conclusions?

Reviewer #1: Yes

Reviewer #2: Yes

2. Has the statistical analysis been performed appropriately and rigorously? 

Reviewer #1: Yes

Reviewer #2: Yes

3. Have the authors made all data underlying the findings in their manuscript fully available?

Reviewer #1: Yes

Reviewer #2: Yes

4. Is the manuscript presented in an intelligible fashion and written in standard English?

Reviewer #1: No

Reviewer #2: Yes

5. Review Comments to the Author

Reviewer #1: Comments to Authors:

1. Is the manuscript technically sound, and do the data support the conclusions?

Yes, the technical aspects of the manuscript are technically sound, and the authors stay within the confines of the data when making conclusions.

2. Has the statistical analysis been performed appropriately and rigorously?

Yes, the statistics required were performed appropriately.

3. Have the authors made all data underlying the findings in their manuscript fully available?

Yes, the data is made available to replicate the study.

4. Is the manuscript presented in an intelligible fashion and written in standard English?

No. Currently, the writing makes it difficult to follow the arguments being made, and much of the text lacks fluency. I am sympathetic to any researchers conducting high-quality research in a non-native language (I am assuming that English is not the native language to the authors) – truly, this is an impressive thing to do, and the authors have my utmost respect for this. However, the writing as it currently stands needs large amounts of editing in each section before I would be able to call it “written in standard English” or written in an “intelligible fashion”. Generally, these language issues are simple to fix, but there are so many of them in the manuscript that the issue becomes noticeable and difficult to move past. Additionally, there are numerous times where certain claims are being made, but without any citations to support the claims.

Ultimately, I think this paper has the potential to be accepted, but a considerable amount of time should be spent revising the text to be clearer, with more citations added to justify many of the claims made in the manuscript.

Introduction:

Line 27: Should the scientific name for rats be given?

Line 27 & 32: rats are categorized as "high and low excitability", yet in line 32 they are referred to as "high-excitable". I think that it is more consistent to have these categorizations match one another, e.g., Line 27: "high- and low-excitability" & Line 32 "high-excitability".

Line 47: Change "Previously" to "Previous".

Linr 47: "low and high excitability" should be changed to "high- and low- excitability". "High excitability" (without a hyphen) is correct when "high" is functioning as an adjective modifying "excitability." Ex: "The rats exhibited high excitability." "High-excitability" (with a hyphen) is correct when used as a compound modifier before a noun. Ex: "High-excitability rats were observed in the study."

Line 47: "Previously studies revealed", yet only a single citation has been given at the end of the sentence?

Line 49: Change "high and low-excitable rats" to "high- and low-excitability rats".

Line 50-53: "We also have demonstrated that genetically determined characteristics of rats with high excitability of the nervous system increase the risk of developing a post-stress inflammatory reaction, as evidenced by both cellular and molecular genetic markers [4, 5]" could be adjusted to something like "Prior research has demonstrated that rats with a genetic predisposition to high excitability of the nervous system results in an increased risk of developing a post-stress inflammatory reaction, ...". However, I believe the following text is either too vague or too low on citations. I see two self-citations, but if there is more research on this topic, then it should be presented to readers.

Line 53-54: Add some citations and explain further. What sort of changes also occurred in low-excitability rats? Have other papers shown this? If so, it is good to cite them to establish that this is published and known within the field.

Line 55: Previous paragraphs should naturally flow to their following paragraph. Moving from molecular genetic markers to gut microbiota is abrupt and confusing. Establish in the prior paragraph that other changes may have an effect on rat stress levels, such as diet (cite) and indeed the gut microbiota itself (cite).

Line 57: remove “noted”. If it was published, then calling it a “note” strikes me as disrespectful to the authors (even if you are one of them!). Perhaps something like “Additionally, prior research has established that the composition of microbiota is …”

Line 59-65: I mean this as gently and as compassionately as possible, but this is too confusing to read.

• Line 59: Change “Since phenotypic differences have cause at the molecular-genetic level of regulation…” to something more like, “Due to the fact that phenotypic differences are heavily regulated in the brain at a genetic and molecular level…”

• Line 60: How did it become relevant? What has prior research already established in mouse brains? Explain the logic in the text. There are zero citations here.

• Line 61-62: This logic is confusing to me. If I understand correctly, you are focusing on the amygdala within the limbic area (side note: should it be “limbic system” and not “limbic area”?) because it plays an important role in emotional and behavioral responses. But aren’t other areas of the limbic system also important? What is the justification for just the amygdala, i.e., what does this particular area provide that other areas of the limbic system don’t?

• The citations are too low. At the very least, line 59 needs citations, and I think more citations in general are required.

Line 63-65: I would like to see what your hypotheses were for the study. I think if you want to submit a manuscript, providing hypotheses can never hurt the manuscript and provides readers with insights regarding what you (the authors) were considering when going into this project.

Regarding the introduction, I think that the paragraphs need to flow better between each other, more explanations and justifications are required (especially in the second half of the introduction), and more citations are required.

Materials and methods:

Line 68: With two groups of n=5, this seems quite low to me. I do worry if outside readers might not take this study seriously because of it.

Line 84-87:

• Line 84-86: citations needed.

• Line 87: All animals were… what?

Line 89: Is there a certification that can be explicitly named? I may be curious to know what certification they possess that allows them to perform the decapitation.

Line 99: I think providing a range of reads is better, and including this basic read information as a cited supplementary table is also valuable. A table like this should include: total reads, total paired reads, read count F, read count R, and so on.

Line 100-101: You write “Read mapping and counting were performed with STAR 2.7.9a”, but what settings did you use? If default, then say so. Also, this needs to be cited.

Line 101: You write “Genome: Rnor_6.0. Annotation: Ensembl v.99. Differential expression: Deseq2 v.1.28.1”, please put more effort into describing what you did. Incorporate the tools you used into the manuscript using proper sentences. Additionally, every one of these programs need to be cited.

Line 107-108: ggplot2 and plotly should both be cited.

Line 109: What software did you use to perform GO? Include any GO analysis software you used and cite it.

Line 110-111: You do not need to type “differentially expressed genes (DEGs)”, because you already have defined this acronym. Simply use “DEGs”.

Line 112: cite clusterProfiler.

Line 112-113: Rattus norvegicus should be italicized.

Line 113: “up- and downregulated” should be “up- and down-regulated”.

Line 115-116: This should be written as a proper sentence.

Line 121: cite STRING.

Results:

Line 136: change “log2fold change ≥|0,38| and padj ≤0,05 (FDR)” to “log2fold change ≥|0.38| and padj ≤0.05 (FDR).

Line 145 (Table 1): I am not sure if this is an artefact of the PDF file, but the spacing on this table is wrong throughout the table.

Additionally, it looks like there are irregular spacings between words. This should be uniform, though again I am not sure if this will appear in the final manuscript. I think this is currently too difficult to edit, because so much of it needs to be fixed.

Line 147-148: I do not think that this is how this figure should be introduced. The text should be introducing and describing what you uncovered first, and then reference the figure number at the end of the (properly written and properly descriptive) sentence.

Line 153: “log2fold change ≥|0,38| and padj ≤0,05 (FDR)” to “log2fold change ≥|0.38| and padj ≤0.05 (FDR)”.

Line 155-156: Remove “Gene Ontology”, you already have defined what GO is. Also, what software was used?

Line 159: “Here and at the Fig.4, 5” is wrong and should be adjusted. Remove “Here” to start. Also, “Fig.4” should be adjusted to Fig. 4”. Re-format and re-write this to be properly descriptive.

Line 161: Why do you write “p.adjust”, and what is the difference between that and “padj” in line 153? Make this consistent. “the dots size - the number of genes” also needs to be re-written to be properly descriptive.

Line 164: You write “The most important processes…”, but “important” is not what these files are describing. Rather, they are describing varying degrees of statistical significance. It is therefore more accurate to write it in terms of statistical significance than perceived importance.

Line 165-166: “regulation of adaptive immune response, “antigen processing and presentation of peptide antigen via MHC class II,” and “regulation of viral entry into host cell”.” is incorrectly written. The commas need to be outside of the closing quotation marks, e.g., “regulation of adaptive immune response”, “antigen processing and presentation of peptide antigen via MHC class II”, and...

Line 167: You have previously written your figures as “Fig 1”, “Fig 2”, “Fig 3”, but now you write “Fig. 4”. This should be adjusted for consistency.

Line 171-172: See my comment above.

Line 175: “MHC protein complex.” Should be “”MHC protein complex”.” – i.e., the periods should be outside of the quotation marks.

Line 195: Remove “identified in our work”. It is already understood that it is your work.

Line 196: Just write GO, not “Gene Ontology (GO)”.

Line 197: “selection criteria: “ please put more effort into describing this. Re-write this to be more descriptive.

Line 198-199: The spacing between paragraphs is inconsistent.

Line 199: “Fig 7 shows that … “You should properly describe the results first, and then insert the figure.

Table 2: “Expression of relevant gene in LT strain compared to the HT strain” has half of the text in bold, while half is not. This should be consistent.

Table 2: Similar to my comment on table 1, the spacing of the text is an issue. As before, I am not sure if this is an artefact of the PDF file, or if editing needs to be done.

Line 212-216: You don’t need to put a summary here. You summarize and contextualize your results in the discussion section.

Discussion:

Line 219: “In actual work” doesn’t mean anything, I would consider removing this entire sentence.

Line 222: Remove “And” from the beginning of the sentence.

Line 228-229: This sounds like future research directions. If you want to include future research directions (which is fine), put it somewhere else, and also properly describe why a lab might do this, and also show what other research has demonstrated already (a quick search online led me to this study: https://pmc.ncbi.nlm.nih.gov/articles/PMC97100/)

Line 229: Remove “On the other hand” and use something like “However”.

Line 233: Are there multiple molecules/orthologs being referenced when you write “CD99 molecule-like 2 - cd99l2”?

Line 236: Replace “It is known” with something like “Prior research has shown that…”. And include more sources if they exist, you only have one citation used in this entire paragraph, but surely there is more research available on this topic?

Line 259-260: This first sentence is too vague, and more sources are needed, if they exist.

Line 272: “Although no studies have been found about the function of Prmt11 within the CNS “. Seems incorrect. For example, here is a manuscript citing (https://pubmed.ncbi.nlm.nih.gov/33460120/)

Line 272-273: “it has been clearly demonstrated that dysregulation of PRMT-family enzymes leads to neurological consequences” – citations are needed.

Line 277: “rat” should be plural.

Line 279-280: “NALCN regulates the resting membrane potential in neurons;” – citations needed here.

Line 301: citations needed.

Line 309-310: The commas need to be outside of the closing quotation.

Line 313: “probably” should be changed to “possibly”, and “associate” should be “associated”.

Line 357: MHC has already been defined.

Line 367: citations needed.

Line 401: “… after psychoemotional stress will reveal the neurogenomic mechanisms of the development…” – How can you be sure this is true? This sounds like a hypothesis to me, not a given fact.

References:

Line 404: Should the references be indented and in a different font size than the rest of the manuscript? If not, then this needs to be adjusted accordingly.

Reviewer #2: Abstract need more information with figures

Number of animals used is low and is the power analysis used before starting the experiment

Methodology lacks reference citation

Table 1 and table 2 looks like a general information

• Without employing functional assays (such as qPCR, protein quantification, or immunological assays) to validate important differentially expressed genes (DEGs), the study only used transcriptome data (such as RNA-Seq).

• Despite the rat genome's comparatively good annotation, some genes or isoforms—especially those pertaining to immune-neural interactions—may still be missing functional annotations. This may restrict how DEGs are interpreted and how they might affect immune system function.

• The study might find DEGs linked to immune function, but it doesn't investigate the fundamental processes that connect immune system control with amygdala excitability. The results might not give a good picture of how these DEGs affect immune-neural connections in the absence of mechanistic knowledge.

6. PLOS authors have the option to publish the peer review history of their article (what does this mean? ). If published, this will include your full peer review and any attached files.

**Do you want your identity to be public for this peer review?** For information about this choice, including consent withdrawal, please see our Privacy Policy .

Reviewer #1: No

Reviewer #2: No

---

## [Author Response · Author response to Decision Letter 1]

20 Mar 2025

Reviewer #1: Comments to Authors:

We would like to express our sincere gratitude for the time and effort you have spent reviewing our manuscript. Your detailed comments have been incredibly helpful in improving the quality of our work. We sincerely appreciate your attention to detail. We have tried to take into account all your comments.

Introduction:

Line 27: Should the scientific name for rats be given? corrected

Line 27 & 32: rats are categorized as "high and low excitability", yet in line 32 they are referred to as "high-excitable". I think that it is more consistent to have these categorizations match one another, e.g., Line 27: "high- and low-excitability" & Line 32 "high-excitability". Yes, in our case, it is more appropriate to use high-excitability / low-excitability rats, as the classification is based on a measured property.

Line 47: Change "Previously" to "Previous". corrected

Linr 47: "low and high excitability" should be changed to "high- and low- excitability". "High excitability" (without a hyphen) is correct when "high" is functioning as an adjective modifying "excitability." Ex: "The rats exhibited high excitability." "High-excitability" (with a hyphen) is correct when used as a compound modifier before a noun. Ex: "High-excitability rats were observed in the study." corrected

Line 47: "Previously studies revealed", yet only a single citation has been given at the end of the sentence? corrected

Line 49: Change "high and low-excitable rats" to "high- and low-excitability rats". corrected

Line 50-53: "We also have demonstrated that genetically determined characteristics of rats with high excitability of the nervous system increase the risk of developing a post-stress inflammatory reaction, as evidenced by both cellular and molecular genetic markers [4, 5]" could be adjusted to something like "Prior research has demonstrated that rats with a genetic predisposition to high excitability of the nervous system results in an increased risk of developing a post-stress inflammatory reaction, ...". However, I believe the following text is either too vague or too low on citations. I see two self-citations, but if there is more research on this topic, then it should be presented to readers.

We genuinely consider your suggested wording to be more appropriate. However, since our study focuses on neuroinflammation in these unique rat strains, no other research on this specific topic has been conducted so far.

Line 53-54: Add some citations and explain further. What sort of changes also occurred in low-excitability rats? Have other papers shown this? If so, it is good to cite them to establish that this is published and known within the field.

We have added brief information on the changes observed in low-excitability rats. However, no other studies have examined gut microbiota in these specific rat strains so far. Currently, a paper is in press where we provide a detailed characterization of the gut microbiota in intact rats of these strains using a larger sample. This will help assess the stability and reproducibility of strain-specific differences in microbial composition.

Line 55: Previous paragraphs should naturally flow to their following paragraph. Moving from molecular genetic markers to gut microbiota is abrupt and confusing. Establish in the prior paragraph that other changes may have an effect on rat stress levels, such as diet (cite) and indeed the gut microbiota itself (cite). corrected

Line 57: remove “noted”. If it was published, then calling it a “note” strikes me as disrespectful to the authors (even if you are one of them!). Perhaps something like “Additionally, prior research has established that the composition of microbiota is …” corrected

Line 59-65: I mean this as gently and as compassionately as possible, but this is too confusing to read.

• Line 59: Change “Since phenotypic differences have cause at the molecular-genetic level of regulation…” to something more like, “Due to the fact that phenotypic differences are heavily regulated in the brain at a genetic and molecular level…”

• Line 60: How did it become relevant? What has prior research already established in mouse brains? Explain the logic in the text. There are zero citations here.

• Line 61-62: This logic is confusing to me. If I understand correctly, you are focusing on the amygdala within the limbic area (side note: should it be “limbic system” and not “limbic area”?) because it plays an important role in emotional and behavioral responses. But aren’t other areas of the limbic system also important? What is the justification for just the amygdala, i.e., what does this particular area provide that other areas of the limbic system don’t?

• The citations are too low. At the very least, line 59 needs citations, and I think more citations in general are required.

We have added the necessary information and included relevant citations. There is no doubt that several structures of the limbic system play a significant role in emotion and behavior regulation. However, given the practical limitations associated with the inability to simultaneously perform transcriptomic analysis of multiple brain structures, we focused on the amygdala first as a key area actively involved in processing emotions and responding to stress. This will allow us to obtain a detailed molecular characterization of one of the most important regulatory centers of the limbic system. Transcriptomic analysis of the hippocampus and prefrontal cortex is also necessary, and this is our immediate plan.

Line 63-65: I would like to see what your hypotheses were for the study. I think if you want to submit a manuscript, providing hypotheses can never hurt the manuscript and provides readers with insights regarding what you (the authors) were considering when going into this project.

Regarding the introduction, I think that the paragraphs need to flow better between each other, more explanations and justifications are required (especially in the second half of the introduction), and more citations are required.

We have taken your recommendations into account and added the study hypothesis and provided additional details in the discussion section. Furthermore, we addressed the fact that the hypothesis was partially confirmed through our findings.

Materials and methods:

Line 68: With two groups of n=5, this seems quite low to me. I do worry if outside readers might not take this study seriously because of it.

We understand your concern regarding the relatively small sample size (n=5 per group). However, RNA-seq is a very expensive method, and such limitations prevent us from using a larger number of animals for analysis. While more samples are always preferable for ensuring more reliable results, statistical methods do allow the use of n=5. Additionally, it is worth noting that RNA-seq studies often use even fewer biological replicates in global gene expression experiments. According to the "Best Practices for RNA Sequencing" from the ENCODE Consortium (https://www.encodeproject.org/documents/cede0cbe-d324-4ce7-ace4-f0c3eddf5972/@@download/attachment/ENCODE%20Best%20Practices%20for%20RNA_v2.pdf), experiments can be conducted with as few as two biological replicates.

Moreover, RNA-seq studies published by other research groups commonly use sample sizes of n=5-6 or even n=3, as seen in the following examples:

• Zhao S, Fung-Leung WP, Bittner A, Ngo K, Liu X. Comparison of RNA-Seq and microarray in transcriptome profiling of activated T cells. PLoS One. 2014 Jan 16;9(1):e78644. doi: 10.1371/journal.pone.0078644. eCollection 2014. ("There were a total of six time points, with two biological replicates per time point.")

• Dias C, Feng J, Sun H, Shao NY, Mazei-Robison MS, Damez-Werno D, Scobie K, Bagot R, LaBonté B, Ribeiro E, Liu X, Kennedy P, Vialou V, Ferguson D, Peña C, Calipari ES, Koo JW, Mouzon E, Ghose S, Tamminga C, Neve R, Shen L, Nestler EJ. β-catenin mediates behavioral resilience through Dicer1/microRNA regulation. Nature. 2014 December 4;516(7529):51–55. doi:10.1038/nature13976. (n=3 was used).

• Nahvi, Roxanna J., et al. "Transcriptome profiles associated with resilience and susceptibility to single prolonged stress in the locus coeruleus and nucleus accumbens in male sprague-dawley rats." Behavioural brain research 439 (2023): 114162. (n=5)

Line 84-87:

• Line 84-86: citations needed. We have added links to experimental work.

• Line 87: All animals were… what? fixed a typo

Line 89: Is there a certification that can be explicitly named? I may be curious to know what certification they possess that allows them to perform the decapitation.

The decapitation procedure was performed by a staff member with certification from a specialized training course on 'Modern Methods of Using Laboratory Rodents in Translational Biomedical Research,' which included training in humane euthanasia techniques. We have clarified the information in the article, specifying that the specialist possesses the necessary skills in humane euthanasia techniques.

Line 99: I think providing a range of reads is better, and including this basic read information as a cited supplementary table is also valuable. A table like this should include: total reads, total paired reads, read count F, read count R, and so on.

We fixed the error, our sequencing was performed using single-end reads (50 bp), not paired-end. Therefore, the distinction between forward (F) and reverse (R) read counts is not applicable in our dataset. Instead, we provide total mapped read counts per sample and additional information in Supplement_2 file.

Line 100-101: You write “Read mapping and counting were performed with STAR 2.7.9a”, but what settings did you use? If default, then say so. Also, this needs to be cited. done

Line 101: You write “Genome: Rnor_6.0. Annotation: Ensembl v.99. Differential expression: Deseq2 v.1.28.1”, please put more effort into describing what you did. Incorporate the tools you used into the manuscript using proper sentences. Additionally, every one of these programs need to be cited. Done, Information has been added to materials and methods.

Line 107-108: ggplot2 and plotly should both be cited. done

Line 109: What software did you use to perform GO? Include any GO analysis software you used and cite it. done

Line 110-111: You do not need to type “differentially expressed genes (DEGs)”, because you already have defined this acronym. Simply use “DEGs”. done

Line 112: cite clusterProfiler. done

Line 112-113: Rattus norvegicus should be italicized. done

Line 113: “up- and downregulated” should be “up- and down-regulated”. done

Line 115-116: This should be written as a proper sentence. done

Line 121: cite STRING. done

Results:

Line 136: change “log2fold change ≥|0,38| and padj ≤0,05 (FDR)” to “log2fold change ≥|0.38| and padj ≤0.05 (FDR). fixed

Line 145 (Table 1): I am not sure if this is an artefact of the PDF file, but the spacing on this table is wrong throughout the table.

Additionally, it looks like there are irregular spacings between words. This should be uniform, though again I am not sure if this will appear in the final manuscript. I think this is currently too difficult to edit, because so much of it needs to be fixed. fixed

Line 147-148: I do not think that this is how this figure should be introduced. The text should be introducing and describing what you uncovered first, and then reference the figure number at the end of the (properly written and properly descriptive) sentence. fixed

Line 153: “log2fold change ≥|0,38| and padj ≤0,05 (FDR)” to “log2fold change ≥|0.38| and padj ≤0.05 (FDR)”. fixed

Line 155-156: Remove “Gene Ontology”, you already have defined what GO is. Also, what software was used? Fixed and we have specified soft in the Methods section

Line 159: “Here and at the Fig.4, 5” is wrong and should be adjusted. Remove “Here” to start. Also, “Fig.4” should be adjusted to Fig. 4”. Re-format and re-write this to be properly descriptive. fixed

Line 161: Why do you write “p.adjust”, and what is the difference between that and “padj” in line 153? Make this consistent. “the dots size - the number of genes” also needs to be re-written to be properly descriptive. fixed

Line 164: You write “The most important processes…”, but “important” is not what these files are describing. Rather, they are describing varying degrees of statistical significance. It is therefore more accurate to write it in terms of statistical significance than perceived importance. fixed

Line 165-166: “regulation of adaptive immune response, “antigen processing and presentation of peptide antigen via MHC class II,” and “regulation of viral entry into host cell”.” is incorrectly written. The commas need to be outside of the closing quotation marks, e.g., “regulation of adaptive immune response”, “antigen processing and presentation of peptide antigen via MHC class II”, and... fixed

Line 167: You have previously written your figures as “Fig 1”, “Fig 2”, “Fig 3”, but now you write “Fig. 4”. This should be adjusted for consistency. fixed

Line 171-172: See my comment above. fixed

Line 175: “MHC protein complex.” Should be “”MHC protein complex”.” – i.e., the periods should be outside of the quotation marks. fixed

Line 195: Remove “identified in our work”. It is already understood that it is your work. fixed

Line 196: Just write GO, not “Gene Ontology (GO)”. fixed

Line 197: “selection criteria: “ please put more effort into describing this. Re-write this to be more descriptive. Fixed, Information has been added to materials and methods.

Line 198-199: The spacing between paragraphs is inconsistent. fixed

Line 199: “Fig 7 shows that … “You should properly describe the results first, and then insert the figure. fixed

Table 2: “Expression of relevant gene in LT strain compared to the HT strain” has half of the text in bold, while half is not. This should be consistent. fixed

Table 2: Similar to my comment on table 1, the spacing of the text is an issue. As before, I am not sure if this is an artefact of the PDF file, or if editing needs to be done. fixed

Line 212-216: You don’t need to put a summary here. You summarize and contextualize your results in the discussion section. fixed

Discussion:

Line 219: “In actual work” doesn’t mean anything, I would consider removing this entire sentence. fixed

Line 222: Remove “And” from the beginning of the sentence. fixed

Line 228-229: This sounds like future research directions. If you want to include future research directions (which is fine), put it somewhere else, and also properly describe why a lab might do this, and also show what other research has demonstrated already (a quick search online led me to this study: https://pmc.ncbi.nlm.nih.gov/articles/PMC97100/)

Thanks for the correct accent. We would like to discuss our results in the context of existing data rather than advancing new hypotheses in this section. In the revised text, our results are discussed in the context of previous studies, and speculative formulations are excluded. Hopefully, this ensures that the section is still an interpretation of our results rather than a direction for future research.

Line 229: Remove “On the other hand” and use something like “However”. fixed

Line 233: Are there multiple molecules/orthologs being referenced when you write “CD99 molecule-like 2 - cd99l2”?

CD99L2 refers to a single membrane protein with a highly conserved extracellular domain between rodents and human. In our RNA-seq data, we identified cd99l2 as a differentially expressed gene, without distinction between potential splice variants. We have clarified this by adding information about its evolution conservativity and adjusting the mention of its abbreviated name for accuracy

Line 236: Replace “It is known” with something like “Prior research has shown that…”. And include more sources if they exist, you only have one citation used in this entire paragraph, but surely there is more research available on this topic?

We have incorporated new information regarding the dual role of CD99L2, based on a recently published study (Kang, 2025). This study highlights its functi

---

## [Decision Letter · Decision Letter 1]

1 Apr 2025

PONE-D-24-49620R1Аmygdala DEGs are associated with the immune system function: a comparative transcriptomic study of high- and low-excitability rat strainsPLOS ONE

Dear Dr. Shalaginova, Thank you for submitting your manuscript to PLOS ONE. After careful consideration, we feel that it has merit but does not fully meet PLOS ONE’s publication criteria as it currently stands. Therefore, we invite you to submit a revised version of the manuscript that addresses the points raised during the review process.

We look forward to receiving your revised manuscript.

Kind regards,

Stephen D. Ginsberg, Ph.D.

Section Editor

PLOS ONE

Journal Requirements:

Additional Editor Comments:

Thank you for resubmitting your work to PLOS ONE. Please make the corrections posed by Reviewer #1 so I can render a decision on this manuscript.

**Comments to the Author**

1. If the authors have adequately addressed your comments raised in a previous round of review and you feel that this manuscript is now acceptable for publication, you may indicate that here to bypass the “Comments to the Author” section, enter your conflict of interest statement in the “Confidential to Editor” section, and submit your "Accept" recommendation.

Reviewer #1: All comments have been addressed

Reviewer #2: All comments have been addressed

2. Is the manuscript technically sound, and do the data support the conclusions?

Reviewer #1: Yes

Reviewer #2: Yes

3. Has the statistical analysis been performed appropriately and rigorously?

Reviewer #1: Yes

Reviewer #2: Yes

4. Have the authors made all data underlying the findings in their manuscript fully available?

Reviewer #1: Yes

Reviewer #2: No

5. Is the manuscript presented in an intelligible fashion and written in standard English?

Reviewer #1: Yes

Reviewer #2: Yes

6. Review Comments to the Author

Reviewer #1: Line 201 -- a random period mark is there, it should be removed.

Line 427-432: This is only 2 sentences, so I do not think it should be made as separate paragraphs.

Reviewer #2: The article entitled "Аmygdala DEGs are associated with the immune system function: a comparative transcriptomic study of high- and low-excitability rat strains" has been well modified and the author addressed all the comments and may be accepted for publication.

7. PLOS authors have the option to publish the peer review history of their article (what does this mean? ). If published, this will include your full peer review and any attached files.

**Do you want your identity to be public for this peer review?**  For information about this choice, including consent withdrawal, please see our Privacy Policy .

Reviewer #1: **Yes: ** Konrad Taube

Reviewer #2: No

---

## [Author Response · Author response to Decision Letter 2]

2 Apr 2025

For Reviewer1:

Thank you for reading the manuscript carefully, we have corrected both technical errors in our text.

---

## [Editor Report · Decision Letter 2]

6 Apr 2025

Аmygdala DEGs are associated with the immune system function: a comparative transcriptomic study of high- and low-excitability rat strains

PONE-D-24-49620R2

Dear Dr. Shalaginova,

We’re pleased to inform you that your manuscript has been judged scientifically suitable for publication and will be formally accepted for publication once it meets all outstanding technical requirements.

Kind regards,

Stephen D. Ginsberg, Ph.D.

Section Editor

PLOS ONE

---

## [Editor Report · Acceptance letter]

PONE-D-24-49620R2

PLOS ONE

Dear Dr. Shalaginova,

I'm pleased to inform you that your manuscript has been deemed suitable for publication in PLOS ONE. Congratulations! Your manuscript is now being handed over to our production team.

Kind regards,

on behalf of

Dr. Stephen D. Ginsberg

Section Editor

PLOS ONE